

# Fighting ability, personality and melanin signalling in free-living Eurasian tree sparrows (*Passer montanus*)

Attila Fülöp[1,2], Zoltán Németh[2], Bianka Kocsis[2], Bettina Deák-Molnár[2], Tímea Bozsoky[2], Gabriella Kőmüves[2] and Zoltán Barta[2]

[1] Juhász-Nagy Pál Doctoral School, University of Debrecen, Debrecen, Hajdú-Bihar, Hungary
[2] MTA-DE Behavioural Ecology Research Group, Department of Evolutionary Zoology and Human Biology, University of Debrecen, Debrecen, Hajdú-Bihar, Hungary

Corresponding author
Attila Fülöp, fafeldolgozo@gmail.com

## ABSTRACT

**Background**. Individuals' access to resources is often decided during dyadic contests the outcome of which is determined by the fighting (or competitive) ability of the participants. Individuals' fighting ability (termed also as resource-holding power or potential, RHP) is usually associated with individual features (*e.g.*, sex, age, body size) and is also frequently signalled through various ornaments like the black throat patch (bib) in many birds. Individual personality is a behavioural attribute often linked to fighting ability as well. Based on earlier studies, however, the relationship between personality and fighting ability is far from being straightforward. While accounting for sex and body size, we studied whether exploratory behaviour, an aspect of personality, predicts fighting ability when competing for food during winter in free-living Eurasian tree sparrows (*Passer montanus*). We also investigated whether the bib can serve as a potential indicator of individual competitiveness in this species.

**Methods**. We captured adult tree sparrows, marked them with a unique combination of colour rings, and collected data about the individuals' sex, body size, bib size and exploratory behaviour. Birds were then released and the agonistic behaviour of the marked individuals was recorded while foraging in groups on bird feeding platforms.

**Results**. The probability of winning a fight, a proxy for fighting ability of individuals, was not related to exploratory behaviour, in either of the sexes. However, bib size was positively related to probability of winning in females, but not in males. Body size was not associated with probability of winning neither in males, nor in females.

**Conclusions**. Our results suggest that, at least in tree sparrows, the outcome of dyadic encounters over food during the non-breeding period are not determined by the exploratory personality of individuals. However, our findings provide further support for a status signalling role of the black bib in tree sparrows, and hint for the first time that bib size might function as a status signal in females as well. Finally, our results do not confirm that body size could serve as an indicator of fighting ability (*i.e.*, RHP) in this species.

## INTRODUCTION

Resources (*e.g.*, food, mate, breeding territory) are typically limited in nature, hence access to them directly influences individuals' fitness (*e.g.*, *Ellis, 1995*; *Majolo et al., 2012*). In order to gain access to limited resources individuals often engage in dyadic agonistic interactions. The outcome of these contests (*i.e.*, winning or losing) is expected to be largely determined by the relative fighting ability of individuals (resource-holding power or potential, RHP; *Parker, 1974*; *Maynard Smith & Parker, 1976*; *Arnott & Elwood, 2009*).

Fighting ability was found to be associated with several individual traits. For instance, in a handful of species chances of winning a fight are higher for males than for females (*e.g.*, *Richner, 1989*; *Briffa & Dallaway, 2007*; but see *Couchoux et al., 2021*), or for adults than for juveniles (*e.g.*, *Richner, 1989*; *Couchoux et al., 2021*). Also, larger individuals usually win more often than smaller ones (*e.g.*, *Richner, 1989*; *Lindström, 1992*; *Briffa, 2008*; *Chamorro-Florescano, Favila & Macías-Ordóñez, 2011*; *Rudin & Briffa, 2011*; *Couchoux et al., 2021*). Also, fighting ability was found to be related to certain individual behavioural characteristics as well, like problem solving capacity (less competitive individuals are better problem-solvers; *e.g.*, *Cole & Quinn, 2012*; *O'Shea, Serrano-Davies & Quinn, 2017*) or personality (see below).

Individual personality is commonly defined as consistent among-individual differences in behaviour across time and/or contexts (*Dall, Houston & McNamara, 2004*; *Réale et al., 2007*) and includes traits as activity, exploration-avoidance, boldness-shyness, sociability or aggressiveness (*Réale et al., 2007*). Personality and fighting ability are proposed to covary due to multiple reasons. From a proximate view, personality and fighting ability can be linked by common physiological mechanisms regulating individual behaviour (*e.g.*, metabolic rate, hormones; *Briffa, Sneddon & Wilson, 2015*). Such a link is suggested by the differences in the individual life-history strategies and related physiological profiles of individuals with different personalities, as proposed by the "pace-of-life syndrome" (POLS) framework (*Réale et al., 2010*). Accordingly, individuals with a fast pace-of-life are expected to have a life-history strategy that favours reproduction at the cost of survival (*i.e.*, live fast), a physiological profile characterized by a high metabolic rate, low hypothalamic–pituitary–adrenal (HPA) axis reactivity and low immune response, and more proactive behaviour (*i.e.*, being more active, exploratory, bold, and aggressive). In contrast, individuals with a slow pace-of-life should have a life-history that promotes survival over reproduction (*i.e.*, live slow), a physiology marked by a low metabolic rate, high HPA axis reactivity and high immune response, and a more reactive behaviour (*i.e.*, being less active and exploratory, shy, and less aggressive) (*Réale et al., 2010*). It has to be noted, though, that empirical evidence for POLS is mixed (*e.g.*, *Royauté et al., 2018*; *Moiron, Laskowski & Niemelä, 2020*), probably because important effects influencing trait correlations have been largely neglected in most of the studies (*e.g.*, sex, ecological conditions; *Hämäläinen et al., 2018*; *Immonen et al., 2018*; *Montiglio et al., 2018*; *Tarka et al., 2018*, reviewed by *Dammhahn et al., 2018*). Nevertheless, support for POLS exists (*e.g.*, *Dhellemmes et al., 2021*), but an overall refinement of the integration of life-history, physiology and personality is needed (*Laskowski, Moiron & Niemelä, 2021*). From an ultimate perspective, personality
differences can be maintained by various evolutionary mechanisms (*e.g.*, differences in state, frequency-dependent selection, environmental variation or non-equilibrium dynamics; *Wolf & Weissing, 2010*). Frequency-dependent selection, for instance, can result in the coexistence of individuals with different personalities, if the fitness benefits of individuals is dependent on the frequency of other personality types in the population, similarly to the different competitive strategies, modelled as Hawk-Dove game, or to the different social foraging strategies modelled as producer-scrounger game (*Wolf & Weissing, 2010*). Previous studies have shown that proactive/reactive individuals often play different frequency-dependent behavioural strategies in different contexts (*e.g.*, producer-scrounger tactics during foraging; *Kurvers et al., 2010*; *Fülöp et al., 2019*). Thus, fighting ability and individual personality are expected to be correlated, on one hand, due to the possibly linked physiological mechanisms of the two behaviours, and on the other hand, by the same evolutionary mechanism maintaining the two behaviours (*Briffa, Sneddon & Wilson, 2015*). However, it is still not entirely clear which personality traits contribute more to the fighting ability of individuals and how they are related.

Numerous studies have investigated the relationship between fighting ability and personality in both invertebrates and vertebrates, but reported contrasting results. For instance, bolder *Actinia equina* sea anemone individuals were more likely to be winners in contests over territory (*Rudin & Briffa, 2012*; *Lane & Briffa, 2017*). Similarly, more exploratory great tits (*Parus major*) were also better capable of monopolizing a food source (*Cole & Quinn, 2012*). In contrast, bolder speckled wood butterflies (*Pararge aegeriahermit*) were not more likely to win contests for sunspot territories (*Kaiser, Merckx & Van Dyck, 2019*); and bolder hermit crabs (*Pagurus bernhardus*) did not have a higher attacker success rate when fighting over shells providing shelter, but less bold individuals were better defenders (*Courtene-Jones & Briffa, 2014*). Furthermore, another study in great tits found no relationship between exploratory behaviour and food resource utilization (*O'Shea, Serrano-Davies & Quinn, 2017*). Overall, as suggested even by this non-exhaustive list of previous findings, the relationship between fighting ability and personality is not always straightforward and appears to be intricate.

Engagement of individuals in competitive interactions, and their outcomes, can be influenced by different intrinsic- and extrinsic factors. For instance, phenotypic (*e.g.*, sex, age, size, personality) differences between participants (see references above) or their fighting skills (*Briffa & Lane, 2017*) can shape contests. Motivation (*Nosil, 2002*), resource value (*Arnott & Elwood, 2007*), previous experience (*e.g.*, through the "winner-loser effect"; *e.g.*, *Stuart-Fox & Johnston, 2005*; *Condon & Lailvaux, 2016*; reviewed by *Hsu & Earley, 2006*; *Rutte, Taborsky & Brinkhof, 2006*) or role asymmetry (*e.g.*, prior resident *vs.* newly arrived individual, resource owner *vs.* intruder; *Snell-Rood & Cristol, 2005*; *Sacchi et al., 2009*; *Chamorro-Florescano, Favila & Macías-Ordóñez, 2011*) can also affect individuals' engagement and success in contests. Contests can also be influenced by context (*e.g.*, social context; *Verbeek, Boon & Drent, 1996*; *Verbeek et al., 1999*). Since fights can be affected by this large array of factors, identification of individual determinants of fighting ability may become challenging, especially in natural populations. For instance, many social species live in fission–fusion societies, meaning that groups regularly split and reunite, on which

occasions group composition and hence social context can change unpredictably. When group characteristics (*e.g.*, size, sex ratio, phenotypic composition) change, many of the factors listed above can change as well.

To minimize the costs of among-individual conflicts, various signalling mechanisms have evolved in many species (*e.g.*, colour traits, behavioural displays, vocalizations), through which individuals can communicate their fighting ability and/or dominance status, towards conspecifics (*Maynard Smith & Harper, 2003*; *Tibbetts, 2013*; *Tibbetts, Pardo-Sanchez & Weise, 2022*). One potential signal associated with status is melanin-based colouration ("badges of status"; *e.g.*, *Rohwer, 1975*; *Järvi & Bakken, 1984*; *Hoi & Griggio, 2008*; *Chaine et al., 2011*; *Rat et al., 2015*; reviewed by *Santos, Scheck & Nakagawa, 2011*). Melanins (eumelanin and pheomelanin) are pigment molecules responsible for the grey, brown and black colouration of the integument. Since melanins are endogenously synthesized, in contrast to carotenoid pigments, which are acquired through the diet, the "honesty" of melanin-based colour signals has frequently been questioned (*e.g.*, *Jawor & Breitwisch, 2003*). Evidence, however, suggests that melanin-based colouration can function as honest signal of individual quality (*e.g.*, *McGraw, 2008*; reviewed by *Guindre-Parker & Love, 2014*). Nevertheless, the mechanisms linking melanin-based colouration and condition can be complex (*e.g.*, *D'Alba et al., 2014*). In some species, such as the Eurasian tree sparrow (*Passer montanus*), melanin-based ornaments are possessed by both sexes (*i.e.*, "mutual ornamentation"; *Kraaijeveld, Kraaijeveld-Smit & Komdeur, 2007*). Interestingly, mutual ornamental traits in different species can have either a similar or different signalling function in the two sexes (similar, *e.g.*, *Rohwer, 1975*; *Järvi & Bakken, 1984*; *Pöysä, 1988*; *Rat et al., 2015*; different, *e.g.*, *Fargallo et al., 2014*; *Matsui et al., 2017*; *Mónus et al., 2017*; *Fülöp et al., 2021*).

The Eurasian tree sparrow is a small-sized, highly gregarious passerine which forages in large flocks (up to 80 individuals at our study site; *Mónus & Barta, 2010*, pers. obs.) during the winter (*Summers-Smith, 1995*; *Barta, Liker & Mónus, 2004*). Although male and female tree sparrows are apparently sexually monochromatic to the naked human eye, a study found evidence for dichromatism in this species in the UV range (*Eaton, 2005*). Sexes are similar in size, though males are slightly larger than females with a large overlap in body size measures of the two sexes (*Mónus et al., 2011*). Both males and females possess a black bib patch (hereafter "bib"), which is a conspicuous eumelanin-based plumage ornament. As a previous study suggests, it can have a sex-dependent status signalling role: it may indicate fighting success in males, but not in females (*Mónus et al., 2017*). Nevertheless, the status signalling role of the bib in females cannot be ruled out, since no other studies, beside the one by *Mónus et al. (2017)*, investigated this question in the two sexes separately. The size of the bib in males is larger than those of females but, similarly to body size measures, there is a considerable overlap between the sexes in this aspect as well (*Mónus et al., 2011*; *Fülöp et al., 2021*). Here, we study whether an axis of individual personality, namely exploratory behaviour, predicts fighting ability in free-living Eurasian tree sparrows in a social foraging context during the non-breeding period (*i.e.,* wintering). We also investigate if fighting ability is communicated towards conspecifics through the individuals' black bib, in other

words, whether the bib has any status signalling role. We study these relationships in the light of the individuals' sex and body size.

## MATERIALS & METHODS

### Study site and period

The study was carried out in the Botanical Garden and the Central Campus of the University of Debrecen (Debrecen, Hungary) during three winters (*i.e.,* between mid-October and mid-March), in 2013–2016. Birds were trapped, marked and tested for exploratory behaviour during all three years of the study, while data on agonistic interactions were collected during the last two years (see also *Fülöp et al., 2019*). The study site is an open area with scattered trees and bushes, also containing some buildings of various sizes forming a heterogeneous semi-urban landscape mosaic. Further details about the study site are given in *Barta, Liker & Mónus (2004)* and *Fülöp et al. (2019)*.

Both the bird trapping procedures and the observations on the social behaviour of individuals were performed on three feeders installed at our study site. Feeders consisted of wooden platforms made of oriented strand board (120 × 120 cm) placed on the ground (*Fülöp et al., 2019*; Fig. 1). During all three winters, we provided *ad libitum* food (sunflower seeds) for the birds on a daily basis, on all the feeders throughout the whole winter period, except periods when recordings of social behaviour were made (see below; *Fülöp et al., 2019*).

### Field procedures

Field procedures were the same as described in *Fülöp et al. (2019)*. We captured tree sparrows with mist nets (Ecotone, Poland; https://www.ecotone.com.pl/) at bird feeders each winter between mid-October and mid-January, except the first winter (2013/14), when birds were captured during the whole winter (*i.e.,* between mid-October and mid-March). At capture, we fitted each individual with a standard ornithological metal ring, issued by the Hungarian Bird Ringing Centre, which was supplemented with a unique combination of three plastic colour rings, enabling the identification of the marked individuals from distance. Furthermore, we measured body mass (± 0.1 g with a Pesola spring balance), tarsus length (± 0.01 mm with a digital caliper) and wing length (± 0.5 mm with a ruler); and because sexes are alike in terms of appearance and size in tree sparrows (*Mónus et al., 2011*), we also took a blood sample (∼50–150 µl) from the brachial vein of the birds to carry out molecular sex determination of individuals (for details, see *Fülöp et al., 2019*; *Fülöp et al., 2021*). To quantify bib size (*i.e.,* area; mm$^2$), we photographed the black bib of the birds on the field and measured bib area subsequently from the photographs with the ImageJ software (ver 1.51i ran on a Linux operating system; https://imagej.nih.gov/ij/index.html). While preparing the photographs, we held the focal individual in a standardized position: the head of the bird facing the camera so that the axis of the beak was perpendicular to the axis of the body and the camera sensor plane. We also included on each photograph a standard reference for length (*i.e.,* ruler or millimetre paper), which was necessary to calibrate unit length during measurements. Measurements were completed in three steps for every individual: first, we calibrated unit length on the photograph using the "set scale"

function of ImageJ, then we traced the outline of the black bib patch with the highest accuracy possible using the "freehand selection" tool, and lastly, we obtained bib area using the "measure" function. All bib size measurements were carried out by the same person (AF). To increase measurement precision of our data, we measured bib size for each individual from the same photo twice and averaged the values. Repeatability of the two measurements was high (intraclass correlation coefficient, performed with the "ICC" package for R (*Wolak, Fairbairn & Paulsen, 2012*; R Core Team, 2021): ICC = 0.991, 95% CI = 0.988–0.994, $N = 175$). Finally, each individual was tested for exploration (see below). After completing all the procedures, individuals were released at the site of their capture. Between 2013 and 2016, we individually marked 194 tree sparrows in total (80 males and 114 females) (*Fülöp et al., 2019*).

## Quantifying exploratory behaviour

We measured exploratory behaviour of individuals using the standard open-field test (*Dingemanse et al., 2002*). The tests and further statistical analyses of the data were carried out as detailed in *Fülöp et al. (2019)*. Briefly, we recorded the behaviour of each individual for 10 min in a novel environment (*i.e.,* an empty room with four artificial trees; room size: $3.25 \times 2.55 \times 1.95$ m (L $\times$ W $\times$ H)) using a handheld video camera (Panasonic HC-V510) through a one-way mirror window. On any day, recordings were made by one of two observers (AF or ZN). Then, from each video, we coded a series of events, as follows: the time of the first landing of the focal individual after its release into the test room, the time of its first movement after the first landing, the number and type of movements performed, and its propensity to explore the test room. We distinguished three different movement types: hops (short distance movements performed either on the ground or on the trees without the bird using its wings), flight-assisted hops (short distance movements combining hops that are assisted with usually one wing beat), and continuous flights (longer distance movements, lasting usually a few seconds, where the bird performs typical flight). We characterized the individuals' propensity of exploration of the test room by dividing the test room into eight equal-sized imaginary compartments and counting the number of occurrences of the focal individual in the different compartments of the room. Compartment limits of the test room were marked by reference lines painted on the walls of the room. All videos were analysed by the same person (TB) with a video analysis software ("mwrap"; *Bán et al., 2017*).

Using the events coded from the videos, we first calculated the variables "movement latency" (s; *i.e.,* the time elapsed between the first landing of the individual in the test room and its first movement), cumulative "number of movements" performed during the test (*i.e.,* sum of the three movement types), "number of different compartments visited" (*i.e.,* ranging 1–8) and "total number of position switches between compartments" of the test room (*i.e.,* sum of the position switches between compartments). Then, using these four variables, we performed a principal component analysis (PCA) and extracted predicted values for the first principal component (PC1). PC1 strongly correlated with three of the four behavioural variables measured, the exception being "movement latency"; and explained 65.95% variance, having the following loadings for "number of movements"

during the test, "number of different compartments visited" and "total number of position switches between compartments" of the test room: 0.556, 0.534 and 0.518, respectively. Therefore, we used these predicted values from PC1 to characterize exploratory behaviour of individuals. Note that higher PC1 scores indicate a higher exploratory tendency of individuals. In total, between 2013 and 2016 we tested 189 individuals (78 males and 111 females) for exploration.

In order to confirm that exploratory behaviour is a personality trait in tree sparrows, we have tested a subset of individuals ($N = 48$ individuals, 20 males and 28 females) multiple times for exploration (45 individuals twice and three individuals three times). Exploratory behaviour was significantly repeatable (adjusted repeatability calculated with a linear mixed-effects model with normal distribution, R package "rptR" (*Stoffel, Nakagawa & Schielzeth, 2017*): $r = 0.455$, SE $= 0.105$, 95% CI $= 0.253$ to $0.669$, $P < 0.001$, $P_{perm} = 0.002$; see *Fülöp et al., 2019* for details). For repeatedly measured individuals, we used the exploration score of their first exploration test in the subsequent statistical analyses.

## Assessment of fighting ability

In order to determine individual fighting ability we recorded agonistic social interactions between sparrows while foraging socially on feeders during the last two winters of the study (2014/15 and 2015/16) at two of the feeders (for details see *Fülöp et al., 2019*). During the winter of 2014/15, recordings were made on 17 different days, between 7 January and 11 February 2015, while during the winter of 2015/16, on 29 different days, between 21 January and 4 March 2016. Recording sessions were performed during daytime (between 8 AM and 5 PM; GMT +1), overlapping with the activity period of the tree sparrows during the winter. During the recording sessions, we restricted individuals' access to food to stimulate the emergence of various forms of social interactions between them (*i.e.,* agonistic interactions and producer-scrounger foraging tactic use; *Fülöp et al., 2019*). To do this, we placed a second layer of oriented strand board of matching size on the feeders which contained 121 holes (diameter: 24 mm, depth: 18 mm, distance between adjacent holes: 10 cm) arranged in a 11 by 11 rectangular grid (Fig. 1). Before the start of the video recording sessions we first removed all the food from the feeder, then filled 10 randomly selected holes with one teaspoon of corn grit. During a recording session we typically recorded multiple foraging bouts. A foraging bout started when the first individual arrived at the feeder and ended when the last bird from the flock flew away from it. The recordings were made by one of three observers at a time (AF, ZN and ZB) using a video camera (Panasonic HC-V510) without disturbing birds (for details see *Fülöp et al., 2019*).

Agonistic interactions between pairs of individuals (*i.e.,* dyads) were analysed subsequently from the videos by the same person (GC). Agonistic interactions included various forms of behaviour exhibited by one or both of the participants, such as displacement, displaying, pecking and/or fighting. For each dyadic interaction in which at least one colour marked individual was involved, we noted the identity of the initiator, the winner and the loser. Besides, for each interaction we have calculated the density of the foraging individuals on the feeder by dividing the number of tree sparrow individuals with

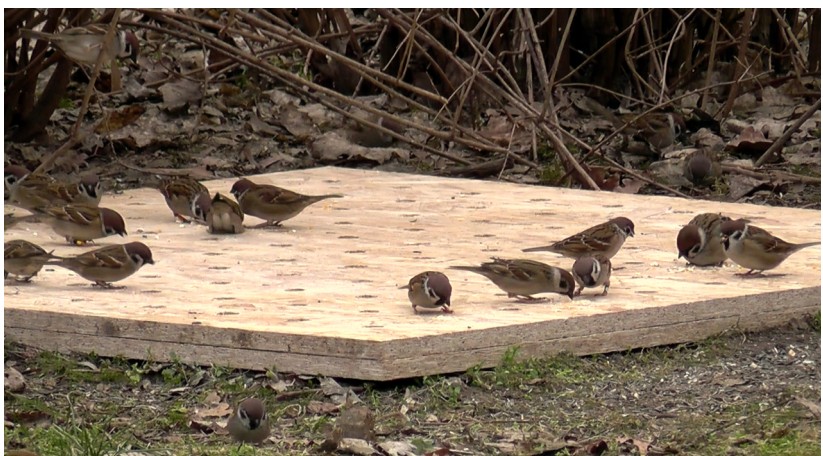

**Figure 1** **Snapshot from a video recording illustrating the feeding platforms used in the study and several Eurasian tree sparrow (*Passer montanus*) individuals, including also colour-ringed birds, in a typical social foraging context.** The feeding platform is made of two layers of oriented strand board, the upper layer containing 121 holes arranged in a 11 by 11 grid (see Methods for details).

the area of the feeder visible in the moment of the interaction from the same video frame (see *Fülöp et al., 2019*). In total, we have recorded a total of $N = 581$ fights, out of which $N = 74$ were fights where no clear winner and loser could be recognized. These latter fights were excluded from the statistical analysis. We were unable to calculate the density of the individuals on the feeder for $N = 25$ fights because the visible area of the feeder could not be determined from the video frame accurately (*e.g.*, the feeder was covered with snow). Hence, these fights were also discarded from the data. Finally, since no exploration data was available for two marked individuals, another $N = 16$ fights were dropped from the data set. Our final data set thus contains $N = 466$ agonistic interactions with an unambiguous outcome from 32 different individuals, of which 14 were males and 18 were females (mean number of fights per individual $= 16.19$, SD $= 21.72$, median $= 6.00$, range $= 1–106$).

We characterised the fighting ability of individuals using the probability of winning as an index. Instead of calculating dominance rank for individuals, we used winning probability because for the majority of fights ($N = 509$ from the total of 581 fights, 87.61%) the identity of only one participant is known and calculating dominance ranks using a data set with a similar structure is not possible (*Sánchez-Tójar, Schroeder & Farine, 2018a*). However, fighting ability (*i.e.*, RHP) is an important individual attribute determining dominance status (*Tibbetts, Pardo-Sanchez & Weise, 2022*), and indeed many previous studies (*e.g.*, *Liker & Barta, 2001*; *Dingemanse & de Goede, 2004*; *Cole & Quinn, 2012*) have shown that fighting ability of individuals is correlated with their dominance status.

## Statistical analyses

All calculations and statistical analyses were performed in the R statistical environment version 4.2.0 (*R Core Team, 2022*) run in R Studio version 2022.02.2.485 (*RStudio Team, 2022*). We have analysed the relationship between individual phenotypic traits and fighting ability with a Bayesian generalized linear mixed-effects model with a binomial response

distribution and logit link function, as implemented in the R package "brms" (*Bürkner, 2017*), an interface for Stan (*Stan Development Team, 2015a*; *Stan Development Team, 2015b*). The model contained fight outcome as a dependent variable (coded with a binary value, 0 or 1, depending whether the individual lost or won the given fight, respectively), observation year (2014/15 and 2015/16) and sex (male or female) as categorical predictors, and the following individual traits as continuous predictors: body size, exploration score and bib size. Agonistic behaviour of individuals, and hence chances of winning fights, can show strong temporal variations on different time scales, as follows. Since tree sparrows usually arrive sequentially on the feeder within a foraging bout, competition between individuals can change (presumably increase) with the time spent on the feeder. Hence, the probability of winning can change within a foraging bout. Furthermore, behaviour of individuals can also vary during the day, due to the daily variation in individuals' energy reserves, and can also change over the season, owing to changes in environmental conditions (*e.g.*, temperature). Therefore, to control for these possible effects, we entered into the model the time of the agonistic event within the foraging bout (expressed in seconds passed from the start of the foraging bout) and time of the foraging bout (expressed as minutes spent from midnight), as continuous predictors. To account for the similarity in conditions within each observation day, instead of including date as an additional continuous effect in the model, we entered date (the unique identity for each observation day) as a random effect. We also accounted for the effects of intraspecific competition and environmental harshness by entering in the model the density of the foraging individuals and the minimum temperature of the day (°C; source of data: https://www.ncei.noaa.gov/access/past-weather/debrecen). In addition to the above listed main effects (*i.e.,* categorical and continuous effects), we have tested in the model all the second order interactions of observation year and sex, with the continuous predictors. Individual identity and foraging bout identity were entered in the model as two additional crossed random effects, to control for repeated measurements from the same individuals and potential temporal autocorrelation in behaviour within the same foraging bout, respectively. Although we carried out our observations on two different feeders, we did not include this effect in our final analysis, since entering feeder identity as a categorical predictor and its' interactions with the continuous variables in the model resulted in convergence difficulties in a preliminary phase of the analysis, due to the moderate sample sizes in terms of number of individuals ($N = 32$ individuals), and due to the uneven distribution of sexes across years and feeders, respectively. For the same reason we omitted also the interaction between observation year and sex from the model. Nevertheless, *a priori* we are not expecting a strong effect of feeder location on the relationship between individual traits and fighting ability, since the two feeders were located close to each other (aprox. 400 m) in the same habitat. Body size was expressed using predicted values of the first principal component (PC1) extracted from a PCA ("prcomp" function in R, package "ade4"; *Dray & Dufour, 2007*) that included body mass, tarsus length and wing length of individuals, all measured at capture (see above). Only PC1, which explained 62.97% of the total variation, had an eigenvalue >1 and it had the following loadings for body mass, tarsus length and wing length, respectively: 0.639, 0.572 and 0.514. Prior to entering into the model, bib size values and time of the agonistic event
within the foraging bout were log-transformed to handle the skewness of these variables, and all the continuous predictors were Z-transformed (mean = 0, SD = 1) using the "scale" function in R to facilitate model convergence. We ran the model with default priors (*i.e.,* relatively uninformative priors), four Hamiltonian Monte Carlo chains, each chain for 12,000 iterations, and using a warm-up period of 2000 iterations. We checked model convergence using trace plots and Rhat values (all Rhat = 1). *Post-hoc* comparisons between different slopes of continuous predictors involved in interactions with a 95% credible interval not overlapping 0 (*i.e.,* effects considered "significant" in a frequentist sense) were performed using the R package "emmeans" (function "emtrends" in R; *Lenth, 2020*). Plots were made using the R package "ggplot2" (function "ggplot" in R; *Wickham, 2016*). We report posterior means for all estimated parameter coefficients ± 95% credibility intervals (CrI) or 95% highest posterior density intervals (HPDI).

## Ethical note

Tree sparrows were ringed under a license from the Hungarian Bird Ringing Center (license nr. 390 accredited to ZB) and permission for the study was granted by the Hajdú-Bihar County Governmental Office, District Office of Debrecen - Department of Environmental and Nature Protection (permit nr. HB/10-KTF/00487-1/2016). The study complies with the European laws regarding animal welfare, and adheres to the ASAB/ABS guidelines for the use of animals in behavioural research.

## RESULTS

Among the individual traits, exploratory behaviour and body size were not related to the probability of winning a fight either alone or in interaction with sex (all 95% CrI overlap 0; Table 1). However, winning probability and bib size were associated in a sex-dependent manner (Table 1; Fig. 2); and *post-hoc* comparisons revealed that females with larger bibs were more likely to win a fight ($\beta = 3.47$, 95% HPDI = 0.61–7.15), whereas in males bib size did not predicted fight outcome ($\beta = -0.75$, 95% HPDI = $-3.88$–2.34).

Probability of winning a fight showed no marked temporal variations at any of the temporal scales considered. Probability of winning was not associated with the time of the agonistic event within the foraging bout in any of the sexes (Table 1). Similarly, time of the foraging bout was not associated with the probability of winning, in males or females (Table 1). Probability of winning was similar in the two years of the study, and the relational patterns between probability of winning and the continuous predictors we have tested were similar in the two years of the study (all 95% CrI of the interactions between observation year and the continuous predictors included 0; Table 1), indicating no strong annual variations in the studied associations.

Probability of winning was sex-dependently influenced by the density of the foraging individuals (Table 1); and *post-hoc* tests indicated that females were less likely to win a fight at high densities ($\beta = -0.57$, 95% HPDI = $-1.14$ to $-0.02$), while in males density did not influenced probability of winning ($\beta = 0.34$, 95% HPDI = $-0.25$–0.95). Minimum temperature of the day had no effect on the probability of winning either alone or in interaction with sex or observation year (Table 1).

**Table 1 Summary of the binomial Bayesian generalized linear mixed-effects model containing the parameter estimates of the predictors of fighting ability (*i.e.*, probability of winning).** Reference levels for the categorical variables "Observation year" and "Sex", are 2014/15 and males, respectively. Therefore, the reported estimates show the extent to which winter of 2015/16 and females differ from winter of 2014/15 and males, respectively. The sign of estimates indicates the direction of associations. Effects for which the 95% credible interval (CrI) does not overlap zero are highlighted in bold. $R^2$ for the Bayesian regression model is 0.42.

**Group-level effects:**

| | Estimate (SE) | 95% CrI |
|---|---|---|
| **Individual identity** | **2.97 (0.99)** | **1.53–5.31** |
| **Foraging bout identity** | **1.36 (0.40)** | **0.62–2.19** |
| **Date** | **0.92 (0.42)** | **0.14–1.84** |

**Population-level effects:**

| | Estimate (SE) | 95% CrI |
|---|---|---|
| Intercept | −0.53 (2.03) | −4.67–3.34 |
| Observation year (2015/16) | −0.53 (1.72) | −3.86–2.94 |
| Sex (female) | −3.28 (2.15) | −8.05–0.55 |
| Body size | −0.52 (2.18) | −4.80–3.91 |
| Bib size | 0.08 (2.25) | −4.34–4.61 |
| Exploration | −0.59 (1.89) | −4.57–2.91 |
| Time | −0.56 (0.99) | −2.52–1.39 |
| Time within foraging bout | 0.52 (0.61) | −0.68–1.73 |
| Density | 0.48 (0.48) | −0.45–1.44 |
| Minimum temperature of the day | −0.01 (0.69) | −1.44–1.28 |
| Observation year (2015/16):Body size | 1.94 (1.34) | −0.70–4.60 |
| Observation year (2015/16):Bib area | −1.75 (1.94) | −5.91–1.82 |
| Observation year (2015/16):Exploration | 0.50 (1.67) | −2.59–4.05 |
| Observation year (2015/16):Time | 0.27 (0.99) | −1.71–2.24 |
| Observation year (2015/16):Time within foraging bout | −0.43 (0.60) | −1.63–0.76 |
| Observation year (2015/16):Density | −0.28 (0.48) | −1.24–0.67 |
| Observation year (2015/16):Minimum temperature of the day | −0.70 (0.75) | −2.14–0.84 |
| Sex (female):Body size | −2.10 (2.22) | −6.79–2.10 |
| **Sex (female):Bib size** | **4.48 (2.29)** | **0.57–9.59** |
| Sex (female):Exploration | 1.75 (1.68) | −1.27–5.39 |
| Sex (female):Time | 0.57 (0.38) | −0.16–1.36 |
| Sex (female):Time within foraging bout | 0.30 (0.36) | −0.40–1.00 |
| **Sex (female):Density** | **−0.92 (0.36)** | **−1.65–−0.22** |
| Sex (female):Minimum temperature of the day | 0.61 (0.45) | −0.25–1.51 |

## DISCUSSION

In this study we investigated the relationship between fighting ability (*i.e.*, probability of winning a fight) and an axis of individual personality (*i.e.*, exploratory behaviour) in free-living tree sparrows during the non-breeding period (*i.e.*, wintering) in a social foraging context. We also studied the role of the black bib, an eumelanin-based plumage ornament
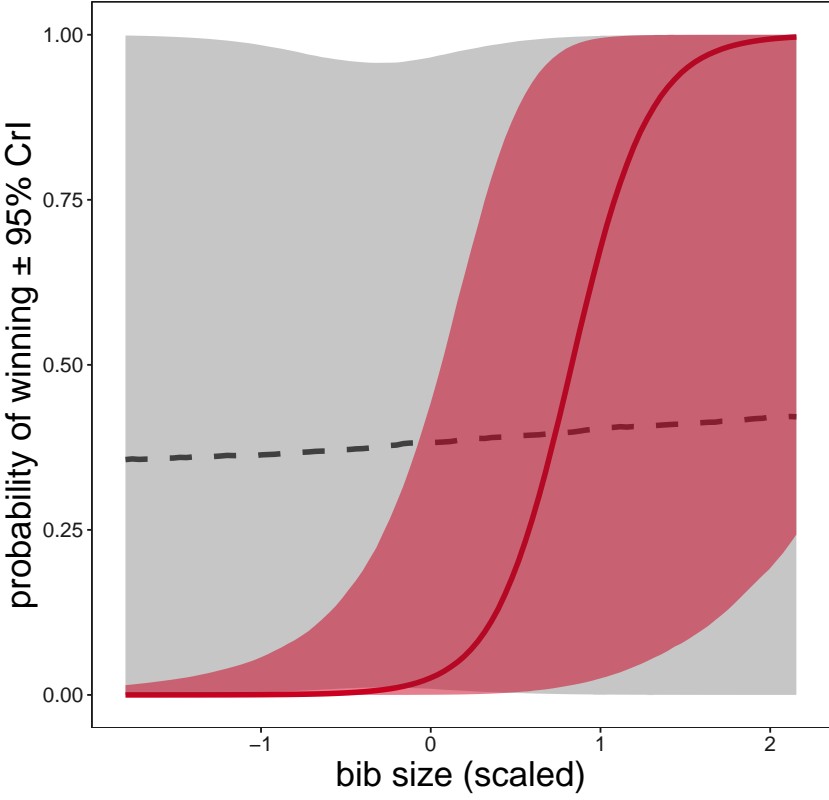

**Figure 2** **Conditional effects plot about the sex-dependent relationship between bib size and probability of winning in free-living Eurasian tree sparrows (*Passer montanus*).** Lines and shaded areas are model predicted regression lines ± 95% credible intervals (CrI), based on values calculated from the binomial Bayesian generalized linear mixed-effects model while all other predictors were held constant apart from the plotted interaction. The grey dashed line indicates males and the pink solid line indicates females.

mutually present in both sexes in this species, as a trait potentially signalling fighting ability. After accounting for the effects of sex and body size, we found that probability of winning was not related to exploratory behaviour in any of the sexes. Bib size, however, was positively associated with probability of winning in females, but not in males. Body size was not associated with probability of winning, neither in males, nor in females.

## Exploratory behaviour and fighting ability

Some of the previous studies concluded that individual personality traits and fighting ability might be related to each other (*e.g.*, *Cole & Quinn, 2012*; *Rudin & Briffa, 2012*; *Lane & Briffa, 2017*), while a handful of others found no support for similar relationships (*e.g.*, *Courtene-Jones & Briffa, 2014*; *O'Shea, Serrano-Davies & Quinn, 2017*; *Kaiser, Merckx & Van Dyck, 2019*). Our results are in line with the latter ones, as we found no association between probability of winning a fight and exploratory behaviour in any of the sexes. This could occur, for instance, if the relationship is context-dependent. Findings supporting context-dependency of this relationship are available from great tits. For example, *Verbeek, Boon & Drent (1996)* and *Verbeek et al. (1999)* found that the sign of relationship between

exploratory behaviour and fighting ability has changed (*i.e.,* positive *vs.* negative) depending on the social context (*i.e.,* individuals tested in small *vs.* large groups). Similarly, *Dingemanse & de Goede (2004)* found that the sign of relationship between exploratory behaviour and dominance, which was positively correlated with the proportion of fights an individual won, varied from positive to negative, depending on the territorial context (*i.e.,* territory holding adult *vs.* non-territorial juvenile individuals). In tree sparrows social context, determined especially by groups' phenotypic composition, might be the most relevant factor influencing trait correlations during the winter. In our study system, group composition seems to be variable both in the short term (*i.e.,* within-season) and in the long term (*i.e.,* between years). In the short term, group composition can vary because tree sparrow flocks likely exhibit a fission–fusion type of group dynamics (pers. obs.). In the long term (*i.e.,* between years), flock membership can change due to emigration/immigration and natural mortality. In both cases flocks' composition might change, potentially reshaping dominance structure of groups and/or determinants of status. Since our study was carried out on free-living individuals, part of them unmarked, we cannot rule out potential confounding effects originating from group dynamics that could influence our results.

A further explanation for not detecting an association between exploratory behaviour and probability of winning may be that the relationship between the two is shaped by a variable we did not measure. Individuals included in the present study can form a mixed sample regarding factors outside the scope of this study, for instance, age, which cannot be determined on the basis of plumage characteristics or other physical traits in tree sparrows. If any such unmeasured trait is important in determining the relationship between exploratory behaviour and fighting ability (see *e.g.,* *Dingemanse & de Goede, 2004*; *Couchoux et al., 2021*), an association between the two could remain masked. Overall, the fact that we have not found a significant relationship between fighting ability and exploratory behaviour does not exclude that a correlation still exists between the two under certain scenarios. In order to clarify these aspects, further studies are needed.

## Sex-specific melanin signalling of fighting ability

The role of conspicuous melanin-based ornaments serving as badges of status has been the focus of multiple studies (reviewed by *Senar, 2006*; *Santos, Scheck & Nakagawa, 2011*), and apparently, the signalling function of various badges of status is far more complex than it was initially expected. This phenomenon is well exemplified by the case of the house sparrow (*Passer domesticus*), where the black bib of males was considered one of the classic examples of status signalling, but the signalling role of the badge has been recently questioned by a meta-analysis (*Sánchez-Tójar et al., 2018b*). Our findings on a closely related, but much less studied species, the tree sparrow, are similarly intriguing. We found that probability of winning a fight was sex-dependently associated with bib size. Moreover, while in a previous study in the same population, performed 15 years earlier (*i.e.,* winter of 1999/2000), *Mónus et al. (2017)* found that bib size was related to the proportion of fights won in males, but not females, here, we found that bib size was associated with a higher probability of winning a fight in females, but not in males. Our results suggest, for the first time, that bib size might have a status signalling function also in females in this species.

The opposite pattern we found regarding the status signalling function of the bib, compared to *Mónus et al. (2017*; see also *Torda, Liker & Barta, 2004*), might be the consequence of the marked changes in environmental factors occurring between the times of the two studies through their effect on or independently of group dynamics (*e.g.*, size, composition, density). Environmental conditions are capable to influence social dynamics for free-living individuals, for instance, through food availability (*e.g.*, quantity, distribution), which can have social implications in the short-term (*i.e.,* within-season) by influencing fission–fusion group dynamics (see *e.g.*, *del Mar Delgado et al., 2021*). In the long-term (*i.e.,* between years), environmental changes can potentially shape population structure (*e.g.*, sex ratio, age structure, size), for instance, through regulating annual survival (*e.g.*, *Santisteban et al., 2012*; *Gullett et al., 2014*). Here, we found that minimum temperature of the day had no effect on the probability of winning, suggesting that individual fighting ability is not influenced by environmental harshness during the winter. In contrast, we found a negative effect of density on the probability of winning in females. This suggests that intraspecific competition can affect fighting ability, at least in one of the sexes. The fact that density apparently acts differently on the fighting ability of individuals in males and females suggests that the RHP of males depends primarily on factors which are independent of their environment (*e.g.*, internal factors), while in females RHP might dependent on internal and/or external factors as well. Nevertheless, the sex-specific effect of density on individual's RHP has to be further studied.

Similarly to exploratory behaviour, another potential explanation for the differences between the two studies can be that one or more confounding effect(s) acting on the association between bib size and fighting ability which could differ between the two studies were not quantified. For instance, body condition, immunity and stress level, or parasite load are among the factors which can be listed as potential candidates. These effects have been shown to be associated with melanin-based colouration (*e.g.*, *Fitze & Richner, 2002*; *Wiebe & Vitousek, 2015*; *Fülöp et al., 2021*; reviewed by *San-Jose & Roulin, 2018*). Although bib size of individuals is acquired during the autumn (*i.e.*, between July and October), when tree sparrows perform their complete annual moult, these factors can still influence the relationship between bib size and fighting ability later in winter, for instance, if bib size is an indicator of individual quality.

Finally, an additional aspect to consider, which is applicable for both studies (*Mónus et al., 2017* study and this study), and which can influence the association between fighting ability and plumage traits both in the short-term (*i.e.,* within-season) and in the long-term (*i.e.,* between years), is familiarity between individuals. Familiarity can be an important factor influencing the information value of plumage signals (*Chaine et al., 2018*). Thus, changes in the social environment can shape the correlation between fighting ability and bib size. We need to point out that for a large fraction of the fights only one of the interacting individuals' identity is known in both studies (see Methods here and in *Mónus et al., 2017*). Hence, limited information is available about the familiarity between opponents in both studies.

It remains an interesting question what mechanism can explain the apparent sex-specific differences in the signalling role of this mutual ornament, as found in multiple instances

in tree sparrows (*e.g.*, *Matsui et al., 2017*; *Mónus et al., 2017*; *Fülöp et al., 2021*, this study), and what are the ultimate consequences of it. Overall, the causes and implications of this phenomenon are poorly understood, and hence, the sex-specific signalling function of mutual colour traits needs to be further investigated.

### Is body size an indicator of fighting ability?

According to theoretical studies (*Parker, 1974*; *Maynard Smith & Parker, 1976*), which are supported also by a large body of empirical evidence (*e.g.*, *Richner, 1989*; *Lindström, 1992*; *Briffa, 2008*; *Chamorro-Florescano, Favila & Macías-Ordóñez, 2011*; *Rudin & Briffa, 2011*; *Couchoux et al., 2021*), larger body size confers a higher RHP to individuals. Therefore, in our analysis we have accounted for the effect of body size, being a potential confounding factor in determining fighting ability of individuals. In contrast with previous expectations, however, body size was not related to the probability of winning a fight in tree sparrows, in any of the sexes.

Interestingly, our results differ again from those of *Mónus et al. (2017)*. They found that a sex-specific relationship exists between measures of body size (*i.e.,* body weight and wing length) and proportion of fights won. Specifically, *Mónus et al. (2017)* showed that in males smaller individuals, while in females larger individuals won more fights. This apparent difference between the two studies might be also the consequence of the marked changes in environmental factors occurring between the times of the two studies and its effect on social dynamics (see above). Similar dynamics occurring in the social environment of individuals might ultimately influence correlational patterns between morphological traits, like body size, and fighting ability, similarly to bib size.

## CONCLUSIONS

Our study contributes to the growing knowledge on the relationship between fighting ability and individual personality. As we found no relationship between exploratory behaviour and probability of winning, our results suggest that in tree sparrows fighting ability of individuals is independent of this personality trait. We found that bib size indicates fighting ability in females, but not in males. Although our results are contrasting with a previous study on the same species on the status signalling function of the black bib (*Mónus et al., 2017*), this finding provides further support for a status signalling role of bib size in this species, and for the first time in females. Finally, we found no relationship between fighting ability and body size—an individual physical trait often associated with competitiveness—in any of the two sexes. Therefore, our results do not support body size as being an indicator of RHP in this species during these years. Overall, taking also into account the correlative approach we used to investigate the above-presented relationships, these thought-provoking findings emphasize the need for further, experimental studies to be carried out in controlled (social) contexts.

## ACKNOWLEDGEMENTS

We thank Mária Borbély, Lilla Buzgó, Réka Csicsek, Angelika Nagy, Katalin Répás and Evelin Szabó for their help in field data collection, and Judit Bereczki and Valéria Mester for their contribution to molecular sexing. We are also grateful to Miklós Bán and Dávid Herczeg for their technical help, and László Papp and the staff of the Botanical Garden for their support. We thank Michael N. Weiss for his advice on the statistical analyses. Maria del Mar Delgado and two anonymous reviewers provided constructive criticism which led to a significant improvement of the manuscript.

### Funding

This work was supported by the National Research, Development and Innovation Office of Hungary (grant number K112527 to Zoltán Barta). Attila Fülöp was supported by a PhD scholarship, by a scholarship from the Magyar Vidékért (Pro Regione) Foundation, and during the preparation of the manuscript, by the Eötvös Scholarship of the Hungarian State (MAEO 2021-22/166661). Zoltán Németh was supported by the National Research, Development and Innovation Office of Hungary (grant number FK124414), the János Bolyai Research Scholarship, and the ÚNKP-21-5 New National Excellence Program of the Ministry for Innovation and Technology from the source of the National Research, Development and Innovation Fund. The work/publication is supported by the EFOP-3.6.1-16-2016-00022 project. The project is co-financed by the European Union and the European Social Fund. The research was supported by the Thematic Excellence Programme (TKP2020-IKA-04) of the Ministry for Innovation and Technology in Hungary. The funders had no role in study design, data collection and analysis, decision to publish, or preparation of the manuscript.

### Grant Disclosures

The following grant information was disclosed by the authors:
The National Research, Development and Innovation Office of Hungary: K112527, FK124414.
The Magyar Vidékért (Pro Regione) Foundation.
The Eötvös Scholarship of the Hungarian State: MAEO 2021-22/166661.
The János Bolyai Research Scholarship.
The ÚNKP-21-5 New National Excellence Program of the Ministry for Innovation and Technology from the source of the National Research, Development and Innovation Fund.
The European Union and the European Social Fund: EFOP-3.6.1-16-2016-00022.
The Thematic Excellence Programme of the Ministry for Innovation and Technology in Hungary: TKP2020-IKA-04.

### Competing Interests

The authors declare there are no competing interests.

## Author Contributions

- Attila Fülöp conceived and designed the experiments, performed the experiments, analyzed the data, prepared figures and/or tables, authored or reviewed drafts of the article, and approved the final draft.
- Zoltán Németh performed the experiments, authored or reviewed drafts of the article, and approved the final draft.
- Bianka Kocsis performed the experiments, authored or reviewed drafts of the article, and approved the final draft.
- Bettina Deák-Molnár performed the experiments, authored or reviewed drafts of the article, and approved the final draft.
- Tímea Bozsoky performed the experiments, authored or reviewed drafts of the article, and approved the final draft.
- Gabriella Kőmüves performed the experiments, authored or reviewed drafts of the article, and approved the final draft.
- Zoltán Barta conceived and designed the experiments, performed the experiments, analyzed the data, authored or reviewed drafts of the article, and approved the final draft.

## Animal Ethics

The following information was supplied relating to ethical approvals (i.e., approving body and any reference numbers):

Hajdú-Bihar County Governmental Office, District Office of Debrecen - Department of Environmental and Nature Protection.

## Data Availability

The data to reproduce the statistical analyses is available in the Supplementary File.

## Supplemental Information

Supplemental information for this article can be found online at http://dx.doi.org/10.7717/peerj.13660#supplemental-information.

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
