# Peer review of "Fighting ability, personality and melanin signalling in free-living Eurasian tree sparrows (Passer montanus)"

_PeerJ, doi:10.7717/peerj.13660_

## Round 0.1 · original submission · Major Revisions

Thank you for your submission. Both reviewers make some excellent suggestions for as to how to improve the manuscript. Please pay attention to each comment. In particular please respond to both reviewers' comments regarding the application and reporting of statistics.

Reviewer 1 ·

Basic reporting

In this well written article, the authors find a link between fighting behaviour and physical traits, as well as a potential "honest signal" function in females, but not males.
I appreciate the thought and effort that went into this experiment to solve a question that to a layman seems easy to solve, but which, after reading the materials and methods is not straightforward.

Most of the relevant literature is present, although an extra ref here and there could be useful (see below). While most remarks are small, I have three remarks that I personally find important.

1)This is probably the shortest results-section I have ever seen. Maybe reconsidered the structure and call it Results and Discussion. An alternative would be to provide some additional summary statistics on exploration (e.g. on average individuals would visit x compartiments) etc.

2) Line 379: is this a likely explanation? How would food shortage promote smaller males being more successful? Have you considered population differences? Age-structural differences (i.e are smaller males young and fit males while bigger birds are old, fat non-fit birds)?

3)Line 408-428: As this is your main results I feel it might be a bit more expanded. I feel that there is a lot of focus on group characteristics influencing the results. However, parasite load, immunity and stress levels could easily differ between populations, explaining the observed results.

Other minor remarks:

Line 61: While it is clear how personality is linked to fighting ability, it is unclear what the “id est” means. Also the “see examples below” can be removed as the examples are listed immediately after.
Detail: the text contains a lot of “like”, to make the text a bit more fluent the author could replace some of the likes with synonyms.
Line 65: I think you can change “were” to “are”.
Line 77: expand a bit on the ultimate perspective. Are these processes really similar, and if so is it then correct to state that they “covary due to multiple reasons” as you mention only one. Even though this has to be clarified a bit, I do appreciate lines 67-77 which are nicely explained.
Line 88/Line 92-95: I would try to find another bird example (in addition to great tits), rather than distantly related species. If such examples do not exist (this is not my expertise), then mention that it is actually quite rare, and rephrase this section that while numerous throughout the animal kingdom, it is rare in birds.
Line 120: this section missed some of the key publications, including McGraw (2008) (“An update on the honesty of melanin-based color signals in birds”). Also, I do not agree with the term “ubiquitous”. Unlike carotenoids (for which ubiquitous could be correct), support for melanin as an honest signal is often uncertain or more complex (see e.g. D’Alba et al. Melanin-based color of plumage: role of condition and feathers’ microstructure). I would simply shift the focus that one potential signal associated with status is melanin-based colouration. Maybe also drop the conspicuous, since melanin is often considered inconspicuous compared to other colours present. Finally, this section is rather short compared to other sections – but this might be just a personal preference.
Figure 1-3, mention R² values in caption.
Line 337: is this not contrasting with Monus et al. 2017 you mention in the introduction?
Line 356: briefly explain what fission-fusion type of group dynamics is.
Line 365: is it possible to measure this if not you can mention this.
Mention in the data accesibility that some data has already been published.

Experimental design

no comment

Validity of the findings

no comment

·

Basic reporting

This manuscript reports the results from an experimental field study that aims to understand whether exploratory behaviour, an aspect of personality, and some other individual features (sex, bib size, etc), explain fighting ability when competing for food during winter in free-living Eurasian tree sparrows. The authors should be commended on a well-conceived and well-designed experiment that, in my opinion, is well balanced and with (at least) some replicates. In addition, I have found the manuscript very well constructed and easy to follow and understand. In particular, the authors have clearly introduced the importance of the study - which I find very interesting indeed - and discussed their results in a broad and general context. I have raised some major and some specific issues below that I hope would improve the manuscript though.
My major comment is about statistics. I think the response variable is a proportion (i.e. number of times an individual win related to the number of encounters). Indeed, this is what the Authors stated in the Abstract (Line 35). However, they analyzed this variable as a binomially distributed one (i.e. the number of fights won vs the number of fights lost) and built a generalized linear model. The number of fights won is in this case related to the number of fights lost because when an individual win a fight, it cannot lose it. In my opinion, it would be more appropriate to estimate the proportion of e.g. fights won. Actually, that was what I thought the Authors did when reading the manuscript. But I got confused by the model they built. This variable should be treated as a beta (that is, a proportion limited between 0 and 1 values) instead of as a binomially distributed one. In addition, some individuals took part in some encounters, but the number of times that the different individuals participated in different fights is different. I think they should include individual as a random factor to account for this unbalanced structure of the data. In this line, the experiment was done in two different years. To account for the potential confounding effect of the year, as the Authors indeed noticed in the Discussion section, year should be also included as a random factor. Finally, I am wondering whether the residuals of their model are temporally autocorrelated, as the experiment was conducted in consecutive (?) days. I would expect the behaviour of the individuals to be more similar on consecutive days than on days that are far away. So my suggestions are:
(1) Build a generalized mixed (see point 2 below) model, but where the residuals of the response variable are beta instead of binomially distributed
(2) individual and year are included as a random factor
(3) check and, if necessary, include a factor to account for the fact that behaviours are temporally autocorrelated
Some specific comments:
Line 77: This is very interesting. Could the Authors expand their explanations about the possible negative frequency-dependent selection?
Line 82: unclear sentence
Line 166: Could the Authors provide the website of the provider?
Line 194: is 10 minutes enough? Has been this time selected based on some previous studies?
Line 375: 15 years earlier but the study was published in 2017. Is this correct, right? Just checking…
Line 378-385: I found it vague to argue that the difference could be only attributed to differences in wintering conditions. There might be many other factors, including experimental design, individual condition, etc
- Could the Authors provide a fit-of-goodness (r-square or deviance explained) of the selected model?

Experimental design

As stated above, I think the experiment is well-conducted. But I need to honestly say here that I am not usually designing and performing experiments. My research is mainly based on observational studies.

Validity of the findings

The authors have clearly introduced the importance of the study - which I find very interesting indeed - and discussed their results in a broad and general context.

Reviewer 3 ·

Basic reporting

.

Experimental design

.

Validity of the findings

.

Additional comments

Attached my comments on the PDF.

Annotated reviews are not available for download in order to protect the identity of reviewers who chose to remain anonymous.

---

## Round 0.2 · Minor Revisions

Thank you for the revised manuscript. Two of the reviewers are happy with the corrections made to the paper. Reviewer 1 still has some outstanding concerns. I would like you to address specifically the question raised by reviewer 1 related to checking for colinearity in your model. Additional rebuttals to the other issues raised in the statistics section would also be helpful. The length of the paper is acceptable for PeerJ. Reviewer 1 attached their comments as an annotated manuscript.

Reviewer 1 ·

Basic reporting

no comment

Experimental design

no comment

Validity of the findings

no comment

Additional comments

no comment

·

Basic reporting

Dear Authors,

I congratulate you on the wonderful revision you made to your manuscript. I really thank you for following my advices on the statistical methods. I think the manuscript has greatly improved its quality. I hope you agree with me. I do not have further comments. I am really impressed by the work you have done on this revision.

Best,
maria

Experimental design

no comment

Validity of the findings

no comment

Additional comments

no comment

Reviewer 3 ·

Basic reporting

Introduction and Discussion overlong compared to content

Experimental design

Reply to previous concerns not satisfactory

Validity of the findings

Reply to previous concerns not satisfactory

Additional comments

The reply to my previous concerns was not entirely satisfactory. Although the authors have somewhat improved the presentation of their results, there is ample room for improvement. I have provided more details in the attached annotated PDF file. I think there are several issues with the analyses that still needs to be addressed, with special reference to contextual effects of foraging group size and temperature. Also, issues of collinearity between sex, body size and bib size were not properly addressed and solved, in my opinion (at least not explicitly).

Annotated reviews are not available for download in order to protect the identity of reviewers who chose to remain anonymous.

---

## Round 0.3 · accepted · Accept

Thank you for your consideration of the reviewers' comments. I am very happy that you have adequately responded to each point in turn. I think the work is very interesting and will make for an impactful paper.